# Examining the application of the IDEAL framework in the reporting and evaluation of innovative invasive procedures: secondary qualitative analysis of a systematic review

Hollie Sarah Richards ![ORCID], Sian Cousins ![ORCID], Darren L Scroggie ![ORCID], Daisy Elliott ![ORCID], Rhiannon Macefield ![ORCID], Elizabeth Hudson, Ian Rodney Mutanga, Maximilian Shah ![ORCID], Natasha Alford, Natalie S Blencowe ![ORCID], Jane Blazeby ![ORCID]

National Institute for Health Research Bristol Biomedical Research Centre Surgical and Orthopaedic Innovation Theme, Bristol Centre for Surgical Research, Bristol Medical School, University of Bristol Medical School, Bristol, UK

**Correspondence to**
Hollie Sarah Richards;
hollie.richards@bristol.ac.uk

## ABSTRACT

**Objectives** The development of new surgical procedures is fundamental to advancing patient care. The Idea, Developments, Exploration, Assessment and Long-term (IDEAL) framework describes study designs for stages of innovation. It can be difficult to apply due to challenges in defining and identifying innovative procedures. This study examined how the IDEAL framework is operationalised in real-world settings; specifically, the types of innovations evaluated using the framework and how authors justify their choice of IDEAL study design.

**Design** Secondary qualitative analysis of a systematic review.

**Data sources** Citation searches (Web of Science and Scopus) identified studies following the IDEAL framework and citing any of the ten key IDEAL/IDEAL_D papers.

**Eligibility criteria** Studies of invasive procedures/devices of any design citing any of the ten key IDEAL/IDEAL_D papers.

**Data extraction and synthesis** All relevant text was extracted. Three frameworks were developed, namely: (1) type of innovation under evaluation; (2) terminology used to describe stage of innovation and (3) reported rationale for IDEAL stage.

**Results** 48 articles were included. 19/48 described entirely new procedures, including those used for the first time in a different clinical context (n=15/48), reported as IDEAL stage 2a (n=8, 53%). Terminology describing stage of innovation was varied, inconsistent and ambiguous and was not defined. Authors justified their choice of IDEAL study design based on limitations in published evidence (n=36) and unknown feasibility and safety (n=32) outcomes.

**Conclusion** Identifying stage of innovation is crucial to inform appropriate study design and governance decisions. Authors' rationale for choice of IDEAL stage related to the existing evidence base or lack of sufficient outcome data for procedures. Stage of innovation was poorly defined with inconsistent descriptions. Further work is needed to develop methods to identify innovation to inform practical application of the IDEAL framework. Defining the concept

---

## STRENGTHS AND LIMITATIONS OF THIS STUDY

⇒ This work provides an in-depth review of how the Idea, Developments, Exploration, Assessment and Long-term (IDEAL) framework has been applied and reported by examining how authors self-report stage of innovation in relation to IDEAL and use this to inform study design decisions.

⇒ There are discrepancies in how the IDEAL framework is operationalised in real-world settings, demonstrating the need for improved methods for identifying innovation in surgery.

⇒ Limitations include the omission of potentially relevant papers where authors did not explicitly reference their use of the IDEAL Framework and that database searches took place in 2020.

of innovation in terms of uncertainty, risk and degree of evidence may help to inform decision-making.

## INTRODUCTION

Innovations in healthcare require evidence of safety and effectiveness before widespread implementation. Although pharmaceutical development occurs through a clearly delineated pathway, surgical innovation often happens haphazardly without incremental building of evidence through evaluation within formal research studies.[1 2] The IDEAL (Idea, Developments, Exploration, Assessment and Long-term follow-up) framework was developed to facilitate the evaluation of new surgical procedures and devices (IDEAL-D).[3 4] It provides study design and governance recommendations for new procedures and devices at different stages of development. Stage 0 deals with preclinical studies, prior to stage 1 first in human studies. Stage 1 is followed by stage 2a



developmental studies, which focus on technical details and ongoing modifications of the intervention, moving to stage 2b exploratory studies where the intervention and patient group are defined in multiple centres prior to proceeding to stage 3 comparative studies. These are typically randomised controlled trials (RCTs) comparing the now stabilised intervention against standard treatment. Stage 4 consists of long-term monitoring in registries.[3 4] Although IDEAL is an important contribution to the field, it has not been widely applied[1 5 6] and is not always used appropriately.[7–9] Challenges to the use of the IDEAL recommendations include that this relies on the user determining the stage of innovation of a procedure a priori, usually by examining the previous literature, and that 'innovation' in itself is not well defined within the IDEAL framework.[10]

Commonly, definitions of surgical innovation are intrinsically linked to concepts of 'newness', including the extent to which a procedure differs from standard practice[11] or is being modified.[11–15] It can also be defined as something with uncertain levels of risk.[16] Requirements for additional training[13 15] and the degree of evidence of safety and efficacy[17 18] have been used to determine how innovative an intervention is. However, some of these definitions are difficult to apply in practice.[19] For example, it is often unclear what degree of modification constitutes innovation.[10 20] Guidance has been developed to help surgeons identify innovative procedures.[21] The Macquarie Tool bases the degree of innovation on surgeons' previous experiences and how much the procedure varies from routine practice. This places the responsibility for identifying innovation on the surgeon,[19] and as such may be open to individual interpretations[19 22] and biases.[23–25] IDEAL suggests that examination of published studies may help determine stage of innovation.[26] Although this is useful for stage 1 proof-of-concept studies, where there is very limited or no published evidence about the innovation, the guidance acknowledges that the literature may be of limited use when interventions are at a later stage of innovation. Distinguishing IDEAL stage 2a from 2b may be problematic as centres achieve stability in a procedure at different times.[26] Given these challenges, the examination of how stage of innovation is determined and used by surgeons is needed.

A systematic review in 2020 examined outcome selection in IDEAL/IDEAL-D studies to inform a core outcome set for new surgical procedures.[27] Following that review, a secondary analysis of identified papers was conducted. The aim of this secondary analysis is to examine how innovative surgical procedures are conceptualised, evaluated and reported using the IDEAL framework and to understand authors' rationale for choice of IDEAL study design. This will inform work to develop practical methods to facilitate determination of stage of innovation, to ensure appropriate study design and governance.[4 20]

## METHODS

A secondary qualitative analysis of papers selected for a previous systematic review[27] was undertaken. Published papers self-reported as IDEAL stage 1–4 studies, reporting evaluations of innovative surgical procedures, were examined using a qualitative framework approach[28] (see online supplemental file 1 for Preferred Reporting Items for Systematic Reviews and Meta-Analyses and the enhancing transparency in reporting the synthesis of qualitative research (ENTREQ) checklists).

### Patient and public involvement

Patients and members of the public were not directly involved in this secondary analysis of a systematic review. However, this work took place within the wider context of the Bristol Biomedical Research Centre Surgical and Orthopaedic Innovation Theme, which has engaged extensively with patient and public involvement and engagement (PPIE) activities. This includes an active PPIE group involved in the co-production of projects and public engagement events about the definitions of new procedures and how they are introduced into clinical practice.

### Searches

Articles identified in a previous systematic review[27] were retrieved. In this review, citation searches of the Web of Science and Scopus databases were undertaken to identify all papers citing any of the ten key IDEAL papers[3 7 26 29–35] published between January 2009 and April 2019 (online supplemental file 2). Records were further filtered to select only those with 'IDEAL' or 'IDEAL-D' in the title or abstract. Searches were updated in June 2020 using the same databases and citation search criteria to identify further studies. Articles were imported into reference management software[36] and duplicates were removed. No further searches were undertaken after 2020 because the current review focused on authors' conceptualisation and description of innovation and justification of their choice of study design. Such factors are unlikely to change across the time since the searches were undertaken.

### Eligibility criteria and definitions

Eligibility criteria for inclusion in the previous review and updated search were studies of invasive procedures and devices of any design. Secondary studies and those not involving living human participants were excluded, along with editorials, commentaries, conference proceedings and letters.[27] Further exclusion criteria for the current review were studies where the intervention under evaluation was not the invasive procedure itself (eg, non-invasive cointerventions). Invasive procedures were defined as interventions where access to the body was gained by incision, natural orifices or endoscopic techniques, or percutaneous puncture where instrumentation is used in addition to the puncture needle.[37] This could include, for example, a diagnostic endoscopy procedure that does not involve an incision. Invasive devices were defined as any

instrument, apparatus or material that is used within or implanted into the body.

## Screening

All studies included in the previous review and update were screened to ensure they met the updated eligibility criteria. Titles and abstracts were reviewed initially by one reviewer (HSR) and ineligible studies were excluded. Full texts of all remaining articles were then screened. At each stage, 20% of articles were screened by a second reviewer (SC). Disagreements were resolved by discussion among the wider study team.

## Data extraction and analysis
### Study characteristics

Study characteristics were extracted using a standardised REDCap electronic data capture form.[38 39] Author-reported IDEAL stage, year of publication, country, the invasive procedure or device under evaluation, number of centres and surgeons, and sample size are reported in the previous systematic review.[27] Additional details about reported oversight (eg, research governance), study design and surgical specialty were extracted by one reviewer (HSR). Study designs, determined by National Institute for Health and Care Excellence definitions,[40] were extracted from all papers by two independent reviewers (HSR and SC). To ensure clinical accuracy, data relating to surgical specialties was double-extracted from 20% of papers by a second independent surgeon reviewer (DLS). Disagreements were resolved by discussion. Characteristics were reported using descriptive statistics, where appropriate.

### Type of surgical innovation under evaluation and rationale for study design

Papers were analysed using a thematic framework approach.[28] This structured approach involves coding data to identify themes to form a framework, which was applied in an iterative and reflexive manner.[28] Full publications were imported into NVivo (V.13) and analysed line by line. Text was coded (whereby text of interest is identified and descriptive labels attached) using a combination of deductive and inductive approaches, allowing for any unexpected themes to develop. This ensured that the analysis was data driven. Three frameworks were developed according to the study aims. The first framework related to the type of innovation under evaluation. All text describing the surgical innovation under evaluation was extracted and coded inductively. For each study, the innovative component under evaluation was identified, for example, if the procedure was described as being used for the first time or if changes to a procedure were reported. Where procedural changes were reported, a typology that facilitates the deconstruction of invasive procedures into their constituent parts was used to identify how many, and which, steps of the procedure were changed.[37] Where codes shared commonalities, these were grouped together into themes and subthemes to

form a framework. Where the innovation under evaluation involved a device or robot, these formed a separate theme. The second framework related to the terminology used to describe the stage of innovation of the procedure. This was analysed according to pre-agreed themes: 'novel/new'—the first or early use of the procedure/device; 'modified'—planned or unplanned changes to the procedure/device or patient group; 'stabilised'—the finalisation of iterative changes or lack of subsequent changes/modifications and 'adopted'—the wider use of the procedure/device in clinical practice. If the terminology describing stage of innovation could not be coded to these themes, additional themes were created. The third framework related to the reported rationale for study design. All text describing the author's reported rationale for choice of study design was extracted and coded inductively. A narrative summary was produced to illustrate findings with reference to study characteristics.

### Framework development

Three multidisciplinary (qualitative research (HSR), health services research (SC) and surgical (DLS)) members of the team independently coded an initial five articles and discussed how the text had been interpreted. This allowed similarities and differences to be incorporated into an initial set of codes and themes with clear definitions.[28] This process was repeated iteratively until agreement was reached on the definitions of codes and themes/subthemes. A senior member of the team (JB) reviewed coded text and the iterative development of frameworks. Each framework was then applied to all articles by one researcher (HSR). The research team (HSR, SC, DC and JB) met regularly throughout this process to discuss analysis, ensuring agreement was reached regarding the appropriate interpretation of the text.

## RESULTS

From the 74 papers retrieved from the previous review[27] and update (online supplemental file 3), 48 were included.

### Study characteristics

Characteristics of included publications are detailed in table 1. The majority of studies were case series (n=36, 75%) or case studies (n=6, 13%). Most studies were reported as IDEAL stage 2a (n=18) although two did not explicitly state a stage: one stated it was stage 1 or 2a[41] and the other self-reported as an early-stage assessment of technical problems during the learning curve 'in line with the IDEAL recommendations'.[42] Nine papers reported the progression of an intervention through multiple IDEAL stages.

Table 2 details reported governance arrangements. 18 (36%) reported research governance, including approval from institutional review boards or research ethics committees overseen by the US Food and Drug Administration or the UK NHS Health Research Authority. Most

**Table 1** Characteristics of included studies

| Study characteristics | | Total, n=48 (%) | Author-reported IDEAL stage | | | | | |
| --- | --- | --- | --- | --- | --- | --- | --- | --- |
| | | | 1 (n=11) | 2a (n=18) | 2b (n=6) | 3 (n=2) | Multiple (n=9) | Not stated (n=2) |
| Study design | Case series | 36 (75) | 7 | 15 | 6 | – | 7 | 1 |
| | Case study | 6 (13) | 4 | 1 | – | – | – | 1 |
| | Cohort study | 4 (8) | – | 2 | – | 1 | 1 | – |
| | Randomised trial | 1 (2) | – | – | – | 1 | – | – |
| | Multiple study designs | 1 (2) | – | – | – | – | 1 | – |
| Surgical specialty | Urology | 25 (52) | 6 | 10 | 4 | 1 | 4 | – |
| | Gastrointestinal surgery | 7 (15) | 1 | 2 | 1 | 1 | 2 | – |
| | Colorectal surgery | 5 (10) | 1 | 2 | – | – | 2 | – |
| | Gynaecology and obstetrics | 2 (4) | 1 | – | 1 | – | – | – |
| | Breast surgery | 3 (6) | – | 2 | – | – | – | 1 |
| | Otolaryngology | 2 (4) | – | 2 | – | – | – | – |
| | Maxillofacial and dental | 2 (4) | 2 | – | – | – | – | – |
| | Orthopaedics | 1 (2) | – | – | – | – | 1 | – |
| | Paediatric | 1 (2) | – | – | – | – | – | 1 |
| Country | European Economic Area (non-UK) | 30 (63) | 6 | 11 | 6 | 1 | 5 | 1 |
| | Other* | 9 (19) | 2 | 3 | – | 1 | 3 | – |
| | UK | 8 (16) | 2 | 4 | – | – | 1 | 1 |
| | North America | 1 (2) | 1 | – | – | – | – | – |
| Publication year | 2011–2015 | 12 (25) | 3 | 4 | – | – | 3 | – |
| | 2016–2020 | 36 (75) | 8 | 14 | 6 | 2 | 6 | – |
| Single or multicentre study | Single centre | 41 (85) | 11 | 15 | 5 | 1 | 7 | 2 |
| | Multi centre | 5 (10) | – | 2 | 1 | 1 | 1 | – |
| | Not reported | 2 (4) | – | 1 | – | – | 1 | – |
| Number of surgeons performing the intervention | 1–2 | 22 (46) | 6 | 10 | – | 2 | 3 | 1 |
| | 3–7 | 5 (10) | – | 2 | 2 | – | 1 | – |
| | Not reported/unclear | 21 (44) | 5 | 6 | 4 | – | 5 | 1 |
| Number of participants | 1–10 | 10 (21) | 7 | 2 | – | – | – | 1 |
| | 11–50 | 26 (54) | 4 | 14 | 2 | – | 5 | 1 |
| | 51–100 | 5 (10) | – | 1 | 1 | – | 3 | – |
| | 101–400 | 5 (10) | – | 1 | 2 | 1 | 1 | – |
| | 401–510 | 2 (4) | – | – | 1 | 1 | – | – |

*These countries were: India (n=2), Turkey, Japan, Sri Lanka, Canada, Southeast Asia, Honk Kong, Sudan.

of these were reported as stage 2a (n=7). 13 (27%) papers reported obtaining local/institutional approval, using a variety of terms such as 'local ethical committee' and 'institutional medical ethical committee' to describe them. Five (10%) papers, including three reported as IDEAL stage 1, stated that ethical approval was not required or obtained. The remaining twelve (25%) papers, including IDEAL stage 1 (n=3) and stage 2a (n=5), did not report any governance approvals.

**Type of innovation under evaluation**

The types of innovations under evaluation were categorised into five themes related to 'newness' (table 3): (1) an entirely new procedure (this included if a procedure had two or more new steps; n=19); (2) one step of the procedure is new; (3) a new device including robotic approach; (4) changes to the sequence of procedure components and (5) changes to the combination of procedure components.

**Table 2** Governance approvals reported by 48 included studies

| Reported governance approvals | | Total (%) N=48 | Author-reported IDEAL stage | | | | | |
|---|---|---|---|---|---|---|---|---|
| | | | 1 (n=11) | 2a (n=18) | 2b (n=6) | 3 (n=2) | Multiple (n=9) | Not stated (n=2) |
| Formal research governance | Institutional review board | 14 | 4 | 5 | 2 | 1 | 2 | – |
| | 'REC ethical approval' | 1 | - | 1 | – | – | – | – |
| | 'REC ethical approval (and NPC)' | 1 | - | 1 | – | – | – | – |
| | 'Regional ethics commissions' | 1 | – | – | – | 1 | – | – |
| | 'National ethics committee' | 1 | – | – | 1 | – | – | – |
| Local or institutional approvals | 'Local ethical committee' | 2 | – | 1 | 1 | – | – | – |
| | 'Local ethics board' | 1 | – | – | – | – | 1 | – |
| | 'Local ethics review committee' | 1 | – | 1 | – | – | – | – |
| | 'Local medical ethical committee' | 2 | – | 1 | 1 | – | – | – |
| | 'Local research ethics committee' | 3 | 1 | 1 | – | – | 1 | – |
| | 'Institution ethical committee' | 1 | – | 1 | – | – | – | – |
| | 'Institutional health sciences research ethics board' | 1 | – | – | – | – | 1 | – |
| | 'Institutional medical ethical committee' | 1 | – | – | – | – | 1 | – |
| | 'Ethical commission of medical school' | 1 | – | – | – | – | 1 | – |
| No ethical approval obtained | 'Current study: no ethical approval' | 1 | – | 1 | – | – | – | – |
| | 'Audit—ethics not required' | 1 | – | – | – | – | 1 | – |
| | 'Ethical approval not required' | 1 | 1 | – | – | – | – | – |
| | 'No further ethics approval required' | 1 | 1 | – | – | – | – | – |
| | 'Not needed due to retrospective design' | 1 | 1 | – | – | – | – | – |
| Not reported | N/A | 12 | 3 | 5 | 1 | – | 1 | 2 |

REC, research ethics committee.

### An entirely new procedure

19 (39%) papers reported an entirely new procedure. Most (n=15) referred to procedures being newly used in a different clinical context, for example, combining existing minimally invasive techniques into a new type of reconstruction or anastomosis in bowel[43] and urological[44] surgery. Of these, eight were classified as IDEAL 1a. Three (6%) other papers described procedures being applied to a different clinical indication, for example, high intensity focused ultrasound applied to focal therapy for prostate tumours.[45]

### One step of the procedure is new

19 other papers described procedures with one-step change. Most (n=8) were stage 2a. 10 involved one step of a procedure being applied to different clinical indication, for example, an existing invasive imaging technique currently used for one type of cancer being used for a different type of cancer.[46–50] Six described changes to access to a body cavity by using minimally invasive techniques instead of open surgery.[41 51–55] One paper described the application of one step of an invasive procedure to a different patient group.[56]

### A new device including robotic approach

Eight (17%) papers described the evaluation of an invasive device/robot. Of these, four described changes to one component of an invasive device (n=1) or robot (n=3). These were a modification to the coating of a vaginal implant device,[57] a new access port for use with a robot devices[58 59] and changing part of the procedure from open to robotic.[60] These were reported as IDEAL stage 1 (n=2), stage 2a (n=1) or stage 3 (n=1), respectively. Three papers described the development of an entirely new invasive device (n=1) or robot (n=2).[61]

### Changes to the sequence of procedure components

One paper reporting a sequential IDEAL stages 2a and 2b study evaluated changes to the timing of radiotherapy for unstable spinal metastases.[62]

### Changes to the combination of procedure components

One IDEAL stage 2b study described an evaluation of combined vs individual invasive diagnostic tests.[63]

### Terminology used to describe stage of innovation

In addition to the four a priori themes, the theme of abandoned procedures was identified during data analyses. The final five themes, (1) new/novel, (2) modified, (3)

**Table 3** Main themes and subthemes describing the type of innovation, by author-reported IDEAL stage and in total

| Main theme | Subthemes (if applicable) | Total (n=48) | Author-reported IDEAL stage and number of papers with text coded to theme | | | | | |
| | | | 1 (n=11) | 2a (n=18) | 2b (n=6) | 3 (n=2) | Multiple stages (n=9) | Not stated (n=2) |
|---|---|---|---|---|---|---|---|---|
| An entirely new procedure | Procedure used for the first time in that clinical context | 15 (31%) | 3 | 8 | – | – | 3 | 1 |
| | Procedure applied to a different clinical indication | 3 (6%) | 1 | – | 1 | – | 1 | – |
| | Procedure previously used in different patient group | 1 (2%) | – | – | 1 | – | – | – |
| One step of the procedure is new | Step of procedure applied to a different clinical indication | 10 (21%) | – | 7 | 2 | – | 1 | – |
| | Change to access | 6 (12%) | 1 | – | 1 | 1 | 2 | 1 |
| | Change to reconstruction | 2 (4%) | 1 | 1 | – | – | – | – |
| | Step of a procedure previously used in different patient group | 1 (2%) | 1 | – | – | – | – | – |
| A new device including robotic approach | Change to one component of a device/robotic instruments | 4 (8%) | 2 | 1 | – | 1 | – | – |
| | Development of an entire device/robotic instruments for one component of procedure | 3 (6%) | 1 | 1 | – | – | 1 | – |
| | Existing device applied to a different clinical indication | 1 (2%) | 1 | – | – | – | – | – |
| Novel changes to the sequence of procedure components | N/A | 1 (2%) | – | – | – | – | 1 | – |
| Changes to the combination of procedure components | N/A | 1 (2%) | – | – | 1 | – | – | – |

IDEAL, Idea, Developments, Exploration, Assessment and Long-term; N/A, not available.

stabilised, (4) adopted and (5) abandoned procedures, are summarised below.

### New/novel

There was a lack of consistency across papers reporting all IDEAL stages in terms describing the 'newness' of the procedure. Commonly terms included 'innovation' (n=18), 'new' (n=14), 'novel' (n=13), 'introduction' (n=4) and 'first-in-human' (n=4). 'Innovation' was used in papers reporting all IDEAL stages, whereas 'introduction' was predominantly used in stage 1 papers (n=3). None of the papers defined these terms. 'First-in-human' was used in one IDEAL stage 1 study, one stage 3 and two multi-stage studies, but in these latter cases, it described the earlier development of the innovation under evaluation. 20 studies used terminology and phrases related to the first report of the innovation (eg, 'this is the first study',[64] 'first clinical trial'[65] and 'the first case'[66]). The majority of these were stage 1 (n=7, 16%) and stage 2a (n=8, 17%). Five papers described the procedure as 'emerging'[41] or 'promising'.[67] These were reported as IDEAL stage 2a (n=3), 2b (n=1) or not stated (n=1).

### Modified

23 papers, mostly early stage 1–2b, used a variety of terms to describe modifications. Although the terms 'modified' or 'modification' were used in 11 (23%) papers, most (n=18, 38%) described modifications in terms of the evolution and ongoing development of the procedure. This included the 'evolution of the new method',[68] describing the procedure as 'still under development'.[69] 10 (21%) papers described changes based on their experience, for example, 'from case 35 onward',[61] 'based on the results of patient 1'[54] and 'thereafter we noticed'.[58] These tended to appear in stage 2a (n=4) and stage 1 (n=3) studies. Some papers used more precise terms, such as 'refinements' (n=7, 16%), 'changes' (n=5, 11%) and 'adaptions/adjustments/alterations' (n=4, 8%). Others were vaguer. Terms included 'hybrid solutions'[58] and 'continued… optimisation'.[70] Additional undefined terms for modifications included 'experimented'[71] and 'improvised',[72] and descriptions of steps of the procedure being 'replaced'[55] or 'tried (and) tested'.[72]

 Richards HS, *et al. BMJ Open* 2024;**14**:e079654. doi:10.1136/bmjopen-2023-079654

**Table 4** Main themes and subthemes describing the reported rationale for author reported study design and IDEAL stage

| Main theme | Subthemes (if applicable) | Total (n=48)* | Author-reported IDEAL stage and number of papers with text coded to theme | | | | | |
|---|---|---|---|---|---|---|---|---|
| | | | 1 (n=11) | 2a (n=18) | 2b (n=6) | 3 (n=2) | Multiple stages (n=9) | Not stated (n=2) |
| Limitations and quality of published evidence | Limited numbers of published studies and small sample sizes | 31 (65%) | 6 | 13 | 5 | 2 | 3 | 2 |
| | Limited outcomes reported | 15 (%) | 3 | 5 | 3 | – | 3 | 1 |
| | Limitations of study designs | 13 (%) | 2 | 6 | 1 | 1 | 2 | 1 |
| | Limitations in quality of previous studies | 4 (8%) | – | 2 | 1 | – | – | 1 |
| | Biases in previous studies | 2 (4%) | 2 | – | – | – | – | – |
| Unknown feasibility and safety outcomes of the procedure | Unknown feasibility outcomes | 25 (%) | 5 | 12 | 2 | – | 4 | 2 |
| | Unknown risk or safety outcomes | 25 (%) | 6 | 10 | 3 | – | 4 | 2 |
| | Study demonstrates proof of concept for use in humans | 2 (4%) | – | – | – | – | 2 | – |
| | Outcomes used to determine progression of IDEAL Stage | 1 (2%) | – | – | – | – | 1 | – |
| Previous use of the procedure in clinical practice | Procedure established in local hospital | 2 (4%) | – | 1 | – | – | 1 | – |
| | Procedure is widely used in clinical practice | 1 (2%) | – | – | 1 | – | – | – |

*Studies may appear in multiple themes.
IDEAL, Idea, Developments, Exploration, Assessment and Long-term.

### Stabilised

Eight papers described the procedure using terms synonymous with stabilised (n=2 stage 1; n=3 stage 2a; n=1 stage 3; n=2 multiple stages). Terms and phrases included having 'achieved stability',[70] 'appear(ing) stable'[69] or 'standardised'.[58] Some authors were slightly more specific and described when procedures were no longer undergoing modifications, in two cases explicitly stating that this was an indication of stabilisation as 'all surgeons began learning' the procedure[51] or it 'had entered IDEAL stage 3'.[55]

### Adopted

Four papers described the procedure being adopted into clinical practice (n=2 stage 2a; n=1 stage 3; n=1 multiple stages). One said the procedure was 'an established technique in our unit'[73] and another said the procedure 'may become a standard'[46] on wider adoption.

### Abandoned

Three papers stated that the procedure had been abandoned (n=1 stage 2a; n=2 multiple stages), including descriptions of the procedure as 'not a viable alternative'[64] or as being halted due to 'loss of confidence'.[51]

### Reported rationale for study design

Three main themes were identified (table 4): (1) the limitations and quality of previously published evidence; (2) unknown feasibility and safety outcomes and (3) previous use in clinical practice.

### The limitations and quality of published evidence

36 papers described the limitations and quality of previously published evidence as a rationale for the current study design. Of these, 31 discussed the limited number of published studies evaluating the procedure. These authors highlighted the small numbers of participants and surgeons in previous studies (eg, 'treated 18 patients'[74] and 'a few expert surgeons'[75]). 15 other papers highlighted limitations in the types of outcomes collected in previous studies and three[74 76 77] (6%) (n=2 stage 2a; n=1 stage 2b) highlighted a lack of long-term outcomes as rationale for the current study. Three papers[44 55 78] (stage 1, stage 2a and stage 2b) stated that the procedure was not standardised. Other papers (n=13) focused on previous study designs and stated that there were few prospective studies and that most evidence was from case series, case–control studies, proof-of-concept or animal model studies. Of these, 10 were reported as stage 2a (n=8) or stage 1 (n=2) studies. In addition, four papers[48 69 75 79] highlighted that findings from previous studies were inconsistent, inconclusive or conflicting, suggesting further evaluation was required. Another two stage 1 papers[54 56] commented on potential biases in previous studies, specifically gender bias, and noted that publication bias had potentially limited unfavourable reports of the procedure under evaluation, justifying their repeat evaluation in a stage 1 study.

### Unknown feasibility and safety outcomes of the procedure

32 papers highlighted unknown feasibility and safety outcomes. 25 specifically discussed feasibility, including 13 that stated establishing technical feasibility was the main aim or primary outcome for the current study. These were IDEAL stage 2a (n=7), stage 1 (n=2), stage 2b (n=2) and multistage (n=2) studies. 25 other papers described unknown risk or safety outcomes, the majority of which were stage 2a (n=10) and stage 1 (n=6). Two[80 81] papers described establishing proof of concept and use in humans, both of which were reported as multiple IDEAL stages, including stage 1. One paper,[51] which reported multiple studies across IDEAL stages, used the emergence of safety outcome data as rationale to progress through the IDEAL stages.

### Previous use of the procedure in clinical practice

Three papers described the extent to which procedure had been used in clinical practice, including being 'an established technique in our unit'[73] or as 'gaining popularity' and as 'increasingly being used…worldwide'.[55] Another paper,[80] reporting multiple IDEAL stages 1–3, described the need to evaluate the procedure against standard practice following its adoption by a second centre.

## DISCUSSION

In this in-depth examination of how the IDEAL framework has been applied in the published literature, we found that most papers reported early IDEAL stage 1 or 2a case series or case studies, reflecting similar findings from other recent literature reviews examining the use of the IDEAL Framework.[8 9] In these papers, reporting of entirely new procedures or those with substantial modifications were included. Governance approvals were varied and inconsistent across IDEAL stages. IDEAL recommendations specify that research ethical approval is required for stages 1–3.[4] However, around one-third of papers did not report seeking any governance approvals, and 13 reported local or institutional approval, the processes and criteria for which are unclear. This has potential implications for patient harm and transparent consent procedures. Terminology used throughout papers varied and was inconsistent. Terms such as 'innovation' and 'modification' were used without supporting definitions. Some papers described the same procedure as both new, modified and established.[73 80] Rationales for the choice of study design were mainly based on published evidence or the need to assess safety or feasibility outcomes, in line with IDEAL recommendations.[4] However, there was some inconsistency with IDEAL recommendations, such as citing the lack of long-term outcomes for the interventions as a reason for evaluating procedures as stage 2a or 2b, despite recommendations that these should measure safety, procedural and short-term outcomes.[4] Clearer, more practically applicable methods are needed to identify the stage of innovation to inform study design and governance decisions.

The IDEAL framework advises that published literature be examined to identify IDEAL stage for an intervention.[26] The inconsistent and ambiguous language used in the published literature to describe stage of innovation highlighted in the current review demonstrates that a clearer approach is required. Others have noted this problem with existing definitions of 'innovation' being described as overly broad[82] and lacking in practical utility because they do not distinguish between expected variation in standard clinical practice and changes made to procedures that are evaluated within research.[24] Ives *et al*[16] recommend a change in terminology is needed around the term innovation. Rather than using words such as 'new, innovative or modified', Ives *et al* recommended that invasive procedures are classified by the level of known risk associated with them based on objective assessments of existing evidence. A new system for categorising published risk for modified or wholly new innovations would allow researchers to map the defined category of risk onto the IDEAL framework to inform study design. The terminology of such a system needs to avoid words that are loaded with inherent bias. The term 'innovation' is synonymous with positive progress and benefit,[16 19] and this may be a factor in influencing research. Positive outcome bias is a recognised limitation in biomedical research.[83–85] This bias may extend to a preference among journal editors to publish more eye-catching studies.[86 87] When introducing or evaluating innovations, instead of a focus on 'how new is this procedure?', a more appropriate question may be 'how does this change affect the procedure in terms of risks and benefits, and subsequent impact on outcomes?'.[16 19] Defining and categorising new and modified interventions in terms of degree of risk and evidence of safety and effectiveness, rather than undefined vague language[16 19] would promote safe and transparent innovation by facilitating the appropriate choice of study design and governance.

In line with recommendations, authors commonly referred to the degree of published evidence as underpinning their choice of study design and IDEAL stage. However, many developments in surgery are not reported in the literature as they are conducted outside of the research framework.[17 88–91] Future work is needed to use methods that provide objective assessments of published literature and additional sources.[92] This may be through systematic application of frameworks such as Grading of Recommendations, Assessment, Development and Evaluations,[93] for developing and summarising evidence, and by ensuring that a body of evidence is built through an incremental approach from case reports and series through to evidence synthesis in systematic reviews. This is essential to ensure patient safety and shared decision-making, share learning and inform evidence-based clinical decision-making.

### Strength and limitations

This review benefits from a robust search and analysis strategy undertaken by a multidisciplinary team of

researchers and clinicians. The screening process may have excluded papers which used the IDEAL framework but did not describe an IDEAL stage in the title or abstract. However, the number of papers identified in this review is comparable to a similar review evaluating the use of the IDEAL framework in study design in the paper published between 2009 and 2017.[8] Our searches took place in June 2020, more than 2 years prior to publication. However, since 2020, the COVID-19 pandemic has impacted the progression of many surgical studies[94–96] and the delivery of surgical interventions.[97 98] As such it is unlikely that the volume of IDEAL publications since our searches took place would be of the levels ordinarily expected within that time frame. We also acknowledge that the introduction of innovative procedures may have been affected by the pandemic and postpandemic changes to clinical practice, which will not be reflected in this data. However, as this review focuses on the methodology, rationale and terminology used within papers self-reporting as IDEAL stage studies, these factors are not as time-dependent as those reviews that examine treatment effects.

## CONCLUSION

This study summarises how the IDEAL framework has been applied in 'real-world' settings for a variety of interventions. It identifies inconsistencies in how authors conceptualise and describe innovation, and how they rationalise their choice of IDEAL stage and study design. Undefined terms were used to describe characteristics of the stage of innovation. It is suggested that innovation in surgery may be better conceptualised in terms of degree of evidence and magnitude of uncertainty about risk and clinical outcomes rather than the nature of the procedure itself. Future work is necessary to develop methods to facilitate this in practical terms and to inform future updates to the IDEAL framework.

**Acknowledgements** The authors acknowledge Christin Hoffman, Shelley Potter and Kerry Avery for their contribution to supervising MS and NA.

**Contributors** JB and NSB conceived the project; RM, EH, IRM, MS and NA developed and performed systematic review searches and screening; HSR, SC, DLS, DE and JB developed the analysis approach and performed analysis. HSR and SC wrote the first draft of the manuscript with substantial input from JB, DLS and DE. HSR, SC, DLS, DE, RM, NSB and JB reviewed and reiterated the final manuscript. JB is the guarantor.

**Funding** This study was funded by the National Institute for Health and Care Research Bristol Biomedical Research Centre.

**Disclaimer** The views expressed are those of the author(s) and not necessarily those of the NIHR or the Department of Health and Social Care.

**Competing interests** None declared.

**Patient and public involvement** Patients and/or the public were not involved in the design, or conduct, or reporting, or dissemination plans of this research.

**Patient consent for publication** Not applicable.

**Provenance and peer review** Not commissioned; externally peer reviewed.

**Data availability statement** Data are available on reasonable request. Data relating to qualitative analysis are available on request.

**ORCID iDs**
Hollie Sarah Richards http://orcid.org/0000-0001-9181-7005
Sian Cousins http://orcid.org/0000-0003-0088-841X
Darren L Scroggie http://orcid.org/0000-0002-5472-2602
Daisy Elliott http://orcid.org/0000-0001-8143-9549
Rhiannon Macefield http://orcid.org/0000-0002-6606-5427
Maximilian Shah http://orcid.org/0000-0001-6522-1945
Natalie S Blencowe http://orcid.org/0000-0002-6111-2175
Jane Blazeby http://orcid.org/0000-0002-3354-3330

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
