## [Reviewer comments · BMJ Open]

ARTICLE DETAILS

TITLE (PROVISIONAL)	Examining the application of the IDEAL Framework in the reporting and evaluation of innovative invasive procedures: secondary qualitative analysis of a systematic review.
AUTHORS	Richards, Hollie; Cousins, Sian; Scroggie, Darren; Elliott, Daisy; Macefield, Rhiannon; Alford, Natasha; Shah, Maximilian; Blencowe, Natalie; Blazeby, Jane

VERSION 1 – REVIEW

REVIEWER	Norman, Gill University of Manchester, Division of Nursing, Midwifery & Social Work
REVIEW RETURNED	13-Oct-2023

GENERAL COMMENTS	I found this an interesting exploration of an important subject. This is a thoughtful and useful approach to a neglected area; which could usefully be drawn upon by other researchers outside the field. I have some comments which the authors may wish to consider. 1) It was not initially clear to me that this is based exclusively on a previous review by the team (albeit with some degree of search update). I think this could be made clearer in the relevant sections of the methods. Having consulted this previously published review I didn't find (and I apologise if I have missed it) any reporting of the full search strategies which would normally be available in supplementary information. It would be good to see these given as supplementary information for this publication. This is particularly important given that only Scopus and Web of Science are reported as searched.2) Use of reporting guidance would be helpful – ENTREQ may be the most appropriate?3) The searches are – as you acknowledge – now over three years old. Even allowing for the impact of the pandemic this feels like a long time. I think the argument that concepts and methods are less susceptible to a developing evidence base than effectiveness is reasonable but only up to a point. This may be particularly the case given that there has been an expansion of research studies using alternative, rapid or truncated methods as a result of the pandemic. It may be important to assess whether there has been a pandemic-influenced development in surgery. I think if it is possible to update the searches it would be worth doing so ahead of publication although I understand that this is a major undertaking.4) This review is looking only at IDEAL stages 1+ so just clarifying explicitly that stage 0 is out of scope would be helpful (this becomes apparent when looking at inclusion criteria)
--

	5) It would also be worth clarifying that invasive procedures potentially include (if I have understood correctly) procedures which are diagnostic in aim and which don't involve use of any incisions etc. 6) I think the suggestions around developments in terms of degrees of evidence and certainty are thoughtful and I would encourage the authors to consider developing them more fully in their discussion – or to set out some possible next steps to achieving their incorporation into a framework.
--	--

REVIEWER	Paez, Arsenio Northeastern University Bouve College of Health Sciences
REVIEW RETURNED	07-Feb-2024

GENERAL COMMENTS	Dear authors, This is a very thought provoking and compelling paper. The results and discussions are well crafted and raise excellent points. My main concerns are with the methods, including whether you can accurately call this a systematic review, and some inconsistencies in the introduction and methods. I think this study can make important contributions to surgical research and innovation in surgery, and applaud your innovative research questions in and potential impact with this paper. I do have several questions and comments, however, but these should be relatively easy to address. The page numbers below refer to original (authors') manuscript pages: Title: I see the reference to the previous systematic review. That makes the title confusing. That means this is a secondary analysis of the data (titles) gathered in that review. Even with updated searches and new eligibility criteria, you are searching within data you previously gathered (papers eligible for a previous systematic review), rather than a new, stand-alone systematic review with independent literature searches and eligible papers. In essence, you are doing a sub-review within a systematic review. I don't think it is appropriate to call this paper a systematic review. The current title implies a new, distinct systematic review. Please edit the title for transparency and accuracy. It may seem like a subtle point, but it does imply something slightly different. page 2 Line 48: suggest innovations (plural), rather than innovation, which would imply something slightly different. Page 3: It is stage 1 that is followed by 2a, but the sentence makes it sound like stage 2a follows stage 0. Line 12: Isn't it more a case that the definition and indications are solidified after being tried with multiple providers and in multiple centres/settings, rather than defined at this stage? These have to be stable and agreed upon before proceeding to stage 3. Line 21-26: This needs a bit of an edit (not quite clear). Is it that the stage of innovation (more clear than innovation stage) for the procedure or device needs to be determined, or is it that the level of evidence to determine the effectiveness or safety of the device or procedure is not yet clear? Those could both be the case, of course, and likely are, but that would be an important distinction or addition to make.
--

	Line 37: What about procedures or devices that are not new, but also don't have high quality evidence for effectiveness, or conflicting evidence about safety? Those would not be new, but rather just poorly evidenced. Lines 41-42: Excellent point! Lines 46-49: These are very important and thought-provoking points. Well done! Page 4: Methods: Please follow the PRISMA guidelines in your description of the methods. It would help with reproducibility and clarity. I see the reference to the previous systematic review. Please see my comments about the title. Page 5 Lines 46-48: The references back to the previous systematic review for key methods and results of searches and screening in this paper further support that this is not an independent, stand-alone systematic review, and should be called something else. It doesn't in any way detract from the fantastic work you have done on an important topic, but is an inadvertently misleading title. Lines 52-54: Please explain why this was done for only surgical specialty? Page 6: References 37 and 38 describe the development and features of Redcap. When reading citation 38, for example, I don't see how this informs your thematic analysis or the iterative and reflexive manner (as opposed to papers or processes such as Saldaña's coding manual and processes, or Braun and Clarke's approach to reflexive thematic analysis, for example). Are these the correct citations for this? Lines 16-17: What was this based on? What model or processes informed this process/inductive and deductive approaches? Lines 34-35: same comment as above, though both of these would be addressed by the same citations. Lines 6-57: All of this would have been guided by some process, model, or framework (for example, Braun and Clarke, Saldaña or others). Please add or cite that. Results: Very interesting and well presented! Page 13 Line 8: It is surprising that only one paper reported this (sequential 2a and 2b). Did you come across other papers with "mixed stages?" I have come across and reviewed paper stating they were a 1 and 2b, 2b or 2a and stage 3, etc, though some of those would have been reviewed and published in 2019, 2020, and between 2020-2023, so they may not have been caught by your searches. It is interesting to consider these in terms of the framework and how adaptable it is, or should be. Discussion: Excellent discussion! Lines 20-22: This is an area in need of further work, development and guidance (from IDEAL as a framework and collaboration). The 2a and 2b studies are, perhaps, the most unique contributions of the framework (which makes very important contributions as a whole), but the stage 2 studies may require more detailed guidance and papers of their own to improve their uptake in the field and the quality of stage 2 studies. Lines 50-53: Excellent point and recommendations (also the idea of risk based terminology in line 37-38) Line 59: risks, and perhaps also benefit (balance of them)? Page 19 Line 46: this is likely true (and a limitation). That raises the question about the "visibility" of the IDEAL framework, or whether the concepts have spread or become diffuse, or whether the
--	---

	concepts are being applied in bits and pieces outside of the framework. Conclusion: lines 22-26: These are very important, thought provoking, and well reasoned recommendations.
--	--

VERSION 1 – AUTHOR RESPONSE

Reviewer: 1

Dr. Gill Norman, University of Manchester

Comments to the Author:

1) It was not initially clear to me that this is based exclusively on a previous review by the team (albeit with some degree of search update). I think this could be made clearer in the relevant sections of the methods. Having consulted this previously published review I didn't find (and I apologise if I have missed it) any reporting of the full search strategies which would normally be available in supplementary information. It would be good to see these given as supplementary information for this publication. This is particularly important given that only Scopus and Web of Science are reported as searched.

Authors response:

Many thanks for highlighting this issue. We have addressed this by amending the title of the manuscript to specify that this is a secondary analysis of a previous systematic review. We have also updated the Methods section to clarify this ("A secondary qualitative analysis of papers selected for a previous systematics review(27) was undertaken.") We have also submitted the citation search strategy details from the initial systematic review as a supplementary file.

2) Use of reporting guidance would be helpful – ENTREQ may be the most appropriate?

Authors response:

Thank you for suggesting this. We have submitted the ENTREQ checklist as supplementary information.

3) The searches are – as you acknowledge – now over three years old. Even allowing for the impact of the pandemic this feels like a long time. I think the argument that concepts and methods are less susceptible to a developing evidence base than effectiveness is reasonable but only up to a point. This may be particularly the case given that there has been an expansion of research studies using alternative, rapid or truncated methods as a result of the pandemic. It may be important to assess whether there has been a pandemic-influenced development in surgery. I think if it is possible to update the searches it would be worth doing so ahead of publication although I understand that this is a major undertaking.

Authors response:

Thank you for highlighting this important point. We agree that ideally an update of the literature would be best. This is not possible however, because staff have moved into different roles. We also think that because this manuscript is a secondary analysis of a previous systematic review looking at a novel issue it does to some extent negate the proposal to repeat the searches. We updated the 'Strength and limitations' sections to reflect your valuable observations around this point:

“We acknowledge that the introduction of innovative procedures may have been affected by the pandemic and post-pandemic changes to clinical practice, which will not be reflected in this data. However, as this systematic review focusses on the methodology, rationale and terminology used within papers self-reporting as IDEAL stage studies, these factors are not as time dependent as those reviews that examine treatment effects.”

4) *This review is looking only at IDEAL stages 1+ so just clarifying explicitly that stage 0 is out of scope would be helpful (this becomes apparent when looking at inclusion criteria)*

Authors response:

Thank you, we have clarified this point in the Methods section.

5) *It would also be worth clarifying that invasive procedures potentially include (if I have understood correctly) procedures which are diagnostic in aim and which don't involve use of any incisions etc.*

Authors response:

Thank you, we have clarified this in the ‘Eligibility criteria and definitions’ section:

“This could include for example a diagnostic endoscopy procedure that does not involve an incision.”

6) *I think the suggestions around developments in terms of degrees of evidence and certainty are thoughtful and I would encourage the authors to consider developing them more fully in their discussion – or to set out some possible next steps to achieving their incorporation into a framework.*

Authors response:

We thank you for these supportive words. The discussion has been expanded upon as below in the first paragraph (final sentence) and by re-drafting the second paragraph.

End of first paragraph

“Clearer, more practically applicable methods are needed to identify the stage of innovation to inform study design and governance decisions, some ideas outlined below.”

Updated second paragraph:

“The IDEAL Framework advises that published literature be examined to identify IDEAL stage for an intervention Pennell 2016(26). The inconsistent and ambiguous language used in the published literature to describe stage of innovation highlighted in the current review demonstrates that a clearer approach is required. Others have noted this problem with existing definitions of ‘innovation’ being described as overly broad Lotz 2013(82) and lacking in practical utility because they do not distinguish between expected variation in standard clinical practice and changes made to procedures that are evaluated within research Rogers 2014(24). Ives *et al* Ives 2022(16) recommend a change in terminology is needed around the term innovation. Rather than using words such as “new, innovative, or modified”, Ives recommended that invasive procedures are classified by the level of known risk associated with them based on objective assessments of existing evidence. A new system for categorising published risk for modified or wholly new innovations would allow researchers to map the defined category of risk onto the IDEAL framework to inform study design. The terminology of such a system needs to avoid words that are loaded with inherent bias. The term ‘innovation’ is synonymous with positive progress and benefit Ives 2022; Birchley 2020(16, 19), and this may be a factor in influencing research. Positive-outcome-bias is a recognised limitation in biomedical research Hasenboehler 2007; Ross 2009; Song 2010(83-85). This bias may extend to a preference amongst journal editors to publish more eye-catching studies Schekman 2013; Viganò 2019(86, 87).

When introducing or evaluating innovations, instead of a focus on 'how new is this procedure?', a more appropriate question may be 'how does this change affect the procedure in terms of risks and benefits, and subsequent impact on outcomes?' Ives 2022; Birchley 2020(16, 19). Defining and categorising new and modified interventions in terms of degree of risk and evidence of safety and effectiveness, rather than undefined vague language Ives 2022; Birchley 2020(16, 19) would promote safe and transparent innovation by facilitating the appropriate choice of study design and governance."

Reviewer: 2

Dr. Arsenio Paez, Northeastern University Bouve College of Health Sciences, University of Oxford
Department of Primary Care Health Sciences

Comments to the Author:

The page numbers below refer to original (authors') manuscript pages:

1. Title: I see the reference to the previous systematic review. That makes the title confusing. That means this a secondary analysis of the data (titles) gathered in that review. Even with updated searches and new eligibility criteria, you are searching within data you previously gathered (papers eligible for a previous systematic review), rather than a new, stand-alone systematic review with independent literature searches and eligible papers. In essence, you are doing a sub-review within a systematic review. I don't think it is appropriate to call this paper a systematic review. The current title implies a new, distinct systematic review. Please edit the title for transparency and accuracy. It may seem like a subtle point, but it does imply something slightly different.

Authors response:

Many thanks for highlighting this issue. We have addressed this by amending the title of the manuscript to specify that this is a secondary analysis of a previous systematic review.

2. Grammar/sentence structure amendments:

page 2

Line 48: suggest innovations (plural), rather than innovation, which would imply something slightly different.

Page 3:

It is stage 1 that is followed by 2a, but the sentence makes it sound like stage 2a follows stage 0.

Line 12: Isn't it more a case that the definition and indications are solidified after being tried with multiple providers and in multiple centres/settings, rather than defined at this stage? These have to be stable and agreed upon before proceeding to stage 3.

Line 21-26: This needs a bit of an edit (not quite clear). Is it that the stage of innovation (more clear than innovation stage) for the procedure or device needs to be determined, or is it that the level of evidence to determine the effectiveness or safety of the device or procedure is not yet clear? Those could both be the case, of course, and likely are, but that would be an important distinction or addition to make.

Line 37: What about procedures or devices that are not new, but also don't have high quality evidence for effectiveness, or conflicting evidence about safety? Those would not be new, but rather just poorly evidenced.

Lines 41-42: Excellent point!

Lines 46-49: These are very important and thought-provoking points. Well done!

Authors response:

Many thanks for these very useful comments. We have updated the manuscript accordingly.

3. *Page 4: Methods: Please follow the PRISMA guidelines in your description of the methods. It would help with reproducibility and clarity. I see the reference to the previous systematic review. Please see my comments about the title.*

Page 5

Lines 46-48: The references back to the previous systematic review for key methods and results of searches and screening in this paper further support that this is not an independent, stand-alone systematic review, and should be called something else. It doesn't in any way detract from the fantastic work you have done on an important topic, but is an inadvertently misleading title.

Authors response:

Many thanks for this. We have updated the title to reflect this and have made amendments to the methods section accordingly. We have also submitted the PRISMA checklist as a supplementary file.

4 *Page 5 Lines 52-54: Please explain why this was done for only surgical specialty?*

Authors response:

Thank you for highlighting this and we apologise for the confusion. We have amended the sentence accordingly – please see below. For clarity, data about research governance, study design and surgical specialty were extracted by one (non-clinical) reviewers. To ensure accuracy, double data extraction was undertaken for text relating to study design (100%) and surgical specialty (20%). The data relating to surgical specialty was double extracted by a clinical member of the team (DS) to ensure non-clinical members were extracting and interpreting the relevant data appropriately. For example, for some procedures it was not always clear to the non-clinical members which surgical specialty (e.g. urology) the procedure fell under.

“Additional details about reported oversight (e.g. research governance), study design, and surgical specialty were extracted by one reviewer (HSR). Study designs, determined by NICE definitions (40) were extracted from all papers by two independent reviewers (HSR, SC). To ensure clinical accuracy, data relating to surgical specialties was double-extracted from 20% of papers by a second independent surgeon reviewer (DS). Disagreements were resolved by discussion. Characteristics were reported using descriptive statistics, where appropriate.”

5. *Page 6: References 37 and 38 describe the development and features of Redcap. When reading citation 38, for example, I don't see how this informs your thematic analysis or the iterative and reflexive manner (as opposed to papers or processes such as Saldaña's coding manual and processes, or Braun and Clarke's approach to reflexive thematic analysis, for example). Are these the correct citations for this?*

Lines 16-17: What was this based on? What model or processes informed this process/inductive and deductive approaches?

Lines 34-35: same comment as above, though both of these would be addressed by the same citations.

Lines 6-57: All of this would have been guided by some process, model, or framework (for example, Braun and Clarke, Saldaña or others). Please add or cite that.

Authors response:

Many thanks for bringing this error in referencing to our attention. The incorrect reference was in place, and we have corrected this. Our analysis was based on the framework analysis methodology

outlined in Gale et al 2013 “Using the framework method for the analysis of qualitative data in multi-disciplinary health research” accessible
here: <https://bmcmmedresmethodol.biomedcentral.com/articles/10.1186/1471-2288-13-117>

6. *Results: Very interesting and well presented!*

Page 13

Line 8: It is surprising that only one paper reported this (sequential 2a and 2b). Did you come across other papers with “mixed stages?” I have come across and reviewed a paper stating they were a 1 and 2b, 2b or 2a and stage 3, etc, though some of those would have been reviewed and published in 2019, 2020, and between 2020-2023, so they may not have been caught by your searches. It is interesting to consider these in terms of the framework and how adaptable it is, or should be.

Authors response:

Thank you for highlighting this. We identified nine papers the reported multiple IDEAL stages, as outlined in Tables 1 and 2. However, we did not report this specific finding in the results section, and as such we have added a sentence to the ‘Study Characteristics’ section:

“Nine papers reported the progression of an intervention through multiple IDEAL stages.”

We have also altered the wording of Line 8 to better reflect this.

6. *Discussion: Excellent discussion!*

Lines 20-22: This is an area in need of further work, development and guidance (from IDEAL as a framework and collaboration). The 2a and 2b studies are, perhaps, the most unique contributions of the framework (which makes very important contributions as a whole), but the stage 2 studies may require more detailed guidance and papers of their own to improve their uptake in the field and the quality of stage 2 studies.

Lines 50-53: Excellent point and recommendations (also the idea of risk based terminology in line 37-38)

Line 59: risks, and perhaps also benefit (balance of them)?

Page 19

Line 46: this is likely true (and a limitation). That raises the question about the “visibility” of the IDEAL framework, or whether the concepts have spread or become diffuse, or whether the concepts are being applied in bits and pieces outside of the framework.

Authors response:

Thank you for your comments and suggestions. We have updated Line 59 to include ‘benefits’ as well as risk.

7. *Conclusion: lines 22-26: These are very important, thought provoking, and well reasoned recommendations.*

Authors response:

Thank you for your useful and insightful comments.

VERSION 2 – REVIEW

REVIEWER	Paez, Arsenio Northeastern University Bouve College of Health Sciences
REVIEW RETURNED	13-Apr-2024
GENERAL COMMENTS	Dear authors, Thank you for these very thoughtful responses and revisions. You raise very thought provoking points in this analysis and I believe it will be of great interest to readers.